# Pharmacokinetic, Metabolomic, and Stability Assessment of Ganoderic Acid H Based Triterpenoid Enriched Fraction of *Ganoderma lucidum* P. Karst

**DOI:** 10.3390/metabo12020097

**Published:** 2022-01-21

**Authors:** Mohd Hafizur Rehman Ansari, Washim Khan, Rabea Parveen, Sadia Saher, Sayeed Ahmad

**Affiliations:** 1Bioactive Natural Product Laboratory, School of Pharmaceutical Education and Research, Jamia Hamdard, New Delhi 110062, India; rehman.ansari1988@gmail.com (M.H.R.A.); khan.washim78@gmail.com (W.K.); rabea_nd62@yahoo.co.in (R.P.); 2Human Genetics Laboratory, Department of Bioscience, Jamia Millia Islamia, New Delhi 110025, India; 3Viral Research Development Laboratory, Department of Microbiology, J.N.M.C, A.M.U, Aligarh 202002, India; sadia.micro@gmail.com

**Keywords:** triterpenoid, ganoderic acid H, quantitative analysis, UPLC–MS, stability

## Abstract

*Ganoderma lucidum* P. karst is an edible fungus that is used in traditional medicine and contains triterpenoids as the major phytoconstituents. Ganoderic acids are the most abundant triterpenoids that showed pharmacological activity. As Indian varieties contain ganoderic acid H (GA-H), we aimed to prepare GA-H-based triterpenoid enriched fraction (TEF) and evaluated its pharmacokinetics, metabolomics, and stability analysis. A high-performance liquid chromatography (HPLC) method was developed to quantify GA-H in TEF and rat plasma. Based on GA-H content, a stability assessment and pharmacokinetic study of TEF were also performed. After its oral administration to rats, TEF’s the metabolic pattern recognition was performed through ultra-performance liquid chromatography mass spectroscopy (UPLC–MS). The developed HPLC method was found to be simple, sensitive, precise (<15%), and accurate (>90% recovery) for the quantification of GA-H. Pharmacokinetic analysis showed that GA-H reached its maximum plasma concentration (C_max_ 2509.9 ng/mL) within two hours and sustained quantifiable amount up to 12 h with a low elimination rate (K_el_) 0.05 L/h. TEF contained ten bioavailable constituents. The prepared TEF was found to be stable for up to one year at room temperature. The prepared TEF, enriched with ganoderic acid, is stable, contains bioavailable constituents, and can be explored as phytopharmaceuticals for different pharmacological properties. Highlights: (1). Preparation of triterpenoid enriched fraction (TEF) from Ganoderma lucidum. (2). Major triterpenoid in TEF is ganoderic acid H (GA-H). (3). TEF contains several bioavailable phytoconstituents. (4). TEF (considering only GA-H) is stable for up to one year at room temperature. (5). GA-H is rapidly absorbed and has high systemic exposure.

## 1. Introduction

The intake of dietary supplements has been increased alarmingly due to several health benefits and trend toward healthy lifestyles. There are numerous herbs and living organisms used as dietary supplements. Fungi are the most widely distributed organisms on earth and are of great medicinal importance. *Ganoderma lucidum*, a fungus known as ‘Mushroom’ in India or ‘Lingzhi’ in China or ‘Reishi’ in Japan, is a well-known medicinal mushroom and has a great future as a dietary supplement. Over 2000 years, *G. lucidum* has been used in several traditional systems as rejuvenation of health by providing vital energy, strengthening the immune system, anti-aging, and strengthening body resistance [1]. Nowadays, it is widely used as a dietary supplement in the United States and in European countries. It is enriched with different classes of secondary metabolites, such as triterpenoids, polysaccharides, peptides, alkaloids, steroids, flavonoids, iridoids, and lignans, and these metabolites have been scientifically proven to have several pharmacological activities [2]. Among these metabolites, triterpenoids are abundant in *G. lucidum*, and these were reported to have hepatoprotective, antioxidant, anti-hypertensive, anti-inflammatory, anti-cholesterol, anticancer, antitumor, anti-HIV-1, neurotrophic, immunomodulatory, and nootropic activities [3]. Since, the first discovery of ganoderic acid GA-A and B, more than 150 triterpenoids have been reported in *G. lucidum* [4,5]. Among them, the major bioactive triterpenoids found in *G. lucidum* are GA-A, GA-B, GA-BS, GA-C1, GA-C2, GA-D, GA-F, GA-H, GA-I, GA-O, GA-J, GA-K, GA-W, GA-M, GA-Ma, GA-DM, GA-X, GA-Y, GA-Z, GA-lactone, GA-δ, GA-θ, GA-η, GA-ζ, GA-α, ganoderiol F, and ganodermanotriol [6].

Major triterpenoids reported in the Chinese variety of *G. lucidum* are GA-A, C2, D, F, and H [2,7]. The ganoderic acid triterpenoids (GA-A, GA-F, and GA-H) of *G. lucidum* were reported to suppress the growth and invasive behavior of breast cancer cells by modulating AP-1 and NF-κB signaling [8]. Indian varieties of *G. lucidum* were reported to have GA-A, D, and F [9,10]. Several in vitro and in vivo studies support the pharmacological activities of Indian varieties of *G. lucidum* [11,12]. However, very few reports have been documented on pharmacokinetic study of *G. lucidum* based on the ganoderic acid A, D, and F content [13,14,15]. In our preliminary study, it was found that GA-H is one of the major triterpenoids found in Indian varieties of *G. lucidum.* Similar to GA-A, GA-H also hydroxylated and converted into bioactive form. GA-H has also exhibited anticancer activity [16]. 

Although other ganoderic acids have been explored, not a single report is available on GA-H. The main objective of this study was to assess *G. lucidum* in terms of GA-H. In the present study, the pharmacokinetics of the triterpenoid-enriched fraction (TEF) of *G. lucidum* were evaluated. Along with the pharmacokinetic study of TEF, we aimed to identify its pattern of metabolites after oral administration in rats through UPLC–MS. This study will lead to the identification of bioavailable compounds present in TEF. Furthermore, GC–MS-based chemical profiling of TEF was performed. Both GC–MS and UPLC–MS have important characteristics to develop effective phytopharmaceutical-based herbal formulations [17]. 

The stability of an herbal formulation is the most prevalent issue for its development as medicine. Bioactive components may undergo oxidation, hydrolysis, and other environmental degradation during natural products’ extraction and development process. The monitoring of the presence and concentration of active constituents is extremely important to check the quality, efficacy, and shelf-life of natural products [18]. To date, no such study has been reported on the stability of TEF. In this context, we have addressed the stability profile of TEF in terms of GA-H content, which will be helpful to develop stable phytopharmaceutical-based herbal products.

## 2. Results

*G. lucidum* mostly contains the lanosterol type of triterpenoids. The solubility criterion (like dissolves like) rule was followed for the extraction–isolation–purification of the TEF from *G. lucidum*. Then, the enrichment of the secondary-metabolites-based extraction process was applied to remove non-terpenoidal impurities. For solvent evaporation, reduced pressure and 50 and 40 °C temperatures were used for 95% ethanol and DCM, respectively, to get dried TEF of *G. lucidum*. The consistency of the produced TEF was syrupy and dark brown in color. The calculated yield of TEF was 0.84% *w*/*w*. TEF was stored in a well-closed container at −20 °C prior to experimental analysis.

### 2.1. Validation of HPLC Method for Quantitative Analysis of GA-H

An in-house high-performance liquid chromatographic (HPLC) method was developed and validated for the quantitative estimation of GA-H. In the developed chromatographic conditions, GA-H was eluted at 17.24 min. The linear concentration range of the GA-H was 100–1000 ng/mL, and the regression equation was Y = 2201.4X + 257,447 with 0.9931 of correlation coefficient (r^2^). The limit of detection (LOD) and limit of quantification (LOQ) of the developed method were 24.71 and 82.39 ng/mL, respectively. The chromatographic conditions were tested for system suitability parameters, such as capacity factor, theoretical plate, height-equivalent theoretical plate, and peak asymmetry at 10% height were 0.14, 8794, 0.019, and 2.15, respectively. The accuracy of the developed method was investigated by testing through the recovery assay. Two different concentrations (500 and 1000 ng/mL) were spiked in two different matrices (plasma and plant extract), followed by extraction and analyzed for its concentration measurement. In both the concentrations, recovery was found as 90.05% and 97.40% with relative standard deviation (%RSD) ≤ 2.0%, indicating good accuracy of the developed method. Besides accuracy, inter-day and intra-day precisions were measured to determine the preciseness of the developed method. For intra-day precision, samples were analyzed in six-replicated within the day. While for inter-day precision, samples were tested on five consecutive days of a week. Both intra-day and inter-day precisions were within the acceptable limits of <15.0% RSD [19]. To determine the robustness of the chromatographic conditions, the developed method was tested for mobile-phase composition, flow rate, and column temperature. Upon slight changes in mobile-phase composition, no significant changes in recovery of GA-H and even no percent change in the detector response of GA-H were observed. It indicates the consistency of the developed method. Details of the method validation are shown in the Appendix A data.

### 2.2. Sample Preparation of GA-H in Plasma for Its Quantitative Estimation

The liquid–liquid extraction method was followed for the extraction of GA-H from plasma. Different solvent systems, such as methanol with formic acid (100:0.5, *v*/*v*); a mixture of methanol, acetonitrile, and formic acid (1:1:0.05, *v*/*v*/*v*); and a mixture of acetonitrile, methanol, and phosphoric acid (1:1:0.05, *v*/*v*/*v*) were tested for optimum extraction of GA-H in plasma. The acceptance criteria for selecting the best extraction solvents were based on the percentage recovery of GA-H and its good precision. Methanol with 0.5% formic acid (*v*/*v*) was found to be the best extracting solvent in which the percentage recovery of GA-H was more than 90% (%RSD ≤ 2%). However, for metabolomic fingerprinting of TEF, the same solvent systems were tested, and we found the maximum number of analytes of interest in the same solvents. The same processed samples were used for pharmacokinetic and metabolomic analysis. 

### 2.3. Pharmacokinetic Analysis of TEF

Pharmacokinetic parameters of TEF in terms of GA-H were calculated from the plasma concentration–time graph (Figure 1). Pharmacokinetic parameters showed that GA-H was rapidly absorbed into the systemic circulation after oral administration of TEF. The C_max_ of GA-H was 2509.9 ± 28.9 ng/mL, peak plasma concentration was achieved at approximately one hour, and AUC_0–t_ and AUC_0–∞_ were 9798.8 ± 169.8 h*ng/mL and 9844.5 ± 157.2 h*ng/mL, respectively. The relative exposure of GA-H was expressed in terms of C_max_/D andAUC_0–∞_/D, and the obtained values were 501.9 ± 25.4 (ng/mL/mg) and 1968.9 ± 32.5 (h*ng/mL/mg), respectively. The volume of distribution (Vd) and clearance (CL) of GA-H were 9660.3 ± 111.5 mL and 507.9 ± 25.6 mL/h, respectively. The half-life of GA-H in blood was 13.18 h, indicating that it was sustained in blood for a longer time and excreted slowly from blood (rate of elimination, K_el_ 0.05 L/h). While mean residence time of GA-H in blood was found to be 5.42 h. In urine analysis, no GA-H was detected. The reason might be that GA-H remained in blood for a longer time, or it might metabolize to other forms. It would be interesting to analyze its metabolites in plasma and urine.

### 2.4. In Vivo Pattern Recognition of TEF

After oral administration of TEF to rats, blood samples were collected and analyzed through UPLC–MS. Compounds were identified by matching masses (m/z) values with different databases. Total nineteen compounds were detected and tentatively identified in TEF after its oral administration to rats (Table 1). In TEF, 14 compounds were identified, out of which only ten compounds were absorbed in systemic circulation after oral administration of TEF to rats. Five different ganoderic acids, namely ganoderic acid N (R_t_ 5.74 min), ganoderic acid D (R_t_ 5.90 min), ganoderic acid A (R_t_ 5.93 min), and ganoderic acid H (R_t_ 7.02 min), were found in TEF, and all of these were bioavailable. However, ganoderic acid H was checked and confirmed by comparing it with the reference standard. Other than ganoderic acids, two more terpenoids, i.e., tsugaric acid A (R_t_ 5.96 min) and hericene (R_t_ 6.12 min), were present in TEF, but these are not bioavailable. One amino acid, i.e., glutamine betaxanthine (R_t_ 4.13 min), was detected in TEF, and it was absorbed in the systemic circulation. In TEF, two bioavailable fatty acids, 13-methylmyristic acid (R_t_ 9.27 min) and linoleic acid (R_t_ 11.24 min), were found. L-carnosine (R_t_ 2.75 min) is the only bioavailable peptide found in TEF. Epicatechin-6-glucoside (R_t_ 3.63 min) was the only bioavailable flavonoid found in TEF. Five endogenous metabolites were detected in UPLC–MS analysis. This UPLC–MS-based metabolomic fingerprinting was rapid and efficient for the identification of targeted and untargeted multi-ingredients.

### 2.5. GC–MS Profiling of TEF of G. Lucidum

GC–MS analysis was carried out for methanolic extract of TEF after derivatization. The GC–MS spectrum confirmed the presence of various components with different retention times as illustrated in Appendix A Appendix A, and the detailed tabulation of most abundant compounds present in TEF are given in Table 2. A total of 49 compounds were separated, out of which 13 compounds were identified through the NIST library. Among identified compounds, the most abundant are fatty acids, followed by terpenoids and some hydrocarbons. However, a list of unknown compounds is given in Appendix A Appendix A. 

### 2.6. Real-Time Stability Profiling of TEF of G. Lucidum

Real-time stability testing is normally performed for a longer and specified duration to check the degradation of active ingredients under recommended storage conditions. During testing, the samples were collected at specific time intervals. In the present study, stability of GA-H was determined in TEF-based samples, and sampling was performed at every three-month interval up to one year (Table 3). No significant change in GA-H content in TEF was observed at both the temperatures (25 and 37 °C) in humid and normal conditions. However, the stability of TEF at 5 °C was also checked, but no significant change in GA-H content was observed.

## 3. Discussion

Over the years, researchers have found that *G. lucidum* is enriched with triterpenoids. Among them, GA-H is one of the abundant triterpenoids found in *G. lucidum.* In this study, TEF was prepared, and GA-H was quantified in it. We used simple liquid–liquid extraction for TEF preparation. Ethanolic extract was suspended in water to dissolve water-soluble compounds, and subsequently DCM was used due to higher affinity of triterpenoids as compared to other phytochemicals. This method is simple and easy to get enriched triterpenoids from crude extract. The quantitative estimation of GA-H from TEF of *G. lucidum* was performed through an in-house validated RP-HPLC method. The RP-HPLC method was simple, precise, sensitive, economical, and best suited for a quantitatively targeted marker-based approach. RP-HPLC analysis of TEF showed a total of 23 peaks in which GA-H eluted at R_t_ 17.26 min and recorded at 254 nm with the most prominent peak. The content of GA-H in TEF was 23.9% (*w*/*w*) of its total dry weight. These results also agree with other studies in which GA-H was found to be the highly abundant triterpenoids in *G. amboinense, G. sessile, G. artum*, and *G. tropicum* [2,7]. In Chinese varieties of *G. lucidum,* GA-A, C2, D, and F were abundant triterpenoids [2,7,20]. However, Indian varieties are also reported to have GA-A, N, D, and F. However, we are the first to report GA-H in Indian varieties of *G. lucidum*. An almost-similar amount of GA-H was found in Indian varieties when compared to other reported varieties. In GC–MS analysis, thirteen different organic compounds, including fatty acids, terpenes, and hydrocarbons, were identified. Thus, the prepared TEF can be explored for the development of GA-H-based phytopharmaceuticals or dietary supplements. The study showed that the *G. lucidum* fruiting-bodies preparations possessed very low toxicity. *G. lucidum* triterpenoids are complex in nature but also safe at a higher dose. These triterpenoids were already well-proven for several biological ailments beyond nutritional benefits [8,21,22]. Pharmacokinetic and bioavailability assessment of triterpenoids of *G. lucidum* have been reported so far with several experimental methods. However, there were no scientific data reported with stable triterpenoid equivalent content with a defined dose. Especially, the pharmacokinetics of TEF, in terms of GA-H content, have not been reported to date. This work identified the GA-H equivalent of TEF as an efficient compound showing a good absorption pattern into the system and studied their pharmacokinetic property in Wistar rats. The GA-H was rapidly absorbed into the blood, i.e., in systemic circulation, after oral administration of TEF. Similar pharmacokinetic profiling was reported for GA-C2 and GA-D [5,15]. The present study indicates good oral bioavailability of TEF in terms of GA-H, and it would be effective in different disorders. With optimized chromatographic conditions, the pattern of metabolites of TEF was identified in blood after its oral administration to rats. It revealed general and metabolomic status of constituents present in TEF up to 72 h. LC–MS-based high-throughput screening strategy of metabolites was easy to access, reliable, and powerful approach for complementary to untargeted profiling [23]. Moreover, previously, no scientific data were reported on pattern of GA-H constituents in blood after oral administration of any kind of extract or fraction. In this study, most of the triterpenoids present in TEF were found to be bioavailable. These bioavailable constituents might show strong biological activities by synergy-based effects [24]. Identification of several triterpenoids of *G. lucidum* was previously reported with best-suited MS-based procedures, but lack of standardization of secondary metabolite-enriched extract and fractions [25].

PCA exhibited the metabolites of TEF and plasma at different time intervals. The cluster of metabolites with significant variation showed two principal components, F1 and F2, in Figure 2A, at a score of 60.41%. From the score plot, samples with a similar pattern of metabolites can be found in the same quadrant, while samples with different metabolite patterns can be found in different quadrants. This exhibited that the metabolites of TEF and plasma at different time intervals (0.5, 24, 48, and 72 h) were in the same quadrant. However, the metabolites at 6 and 12 h were visualized in other quadrants. This concluded that the sample of TEF metabolites was matched with blood. The eigenvalues, cumulative percentage variability, and percentage variance are typically shown in Table 4. The correlation matrix (Pearson value—r) w.r.t pattern of metabolites was calculated (Table 5). From heat-map visualization, metabolites were clearly shown from higher to lower abundance with red, green, and black colors (Figure 2B). We observed that the red-colored metabolites (absorbed in the blood) were showing higher abundance. However, in the case of metabolites of TEF of *G. lucidum,* few metabolites were showing after being absorbed into the blood at different time intervals. The green color of metabolites showed more abundance in the case of sample TEF, but less distributed in subsequent time intervals. This PLS-DA showed a significant and comparative separation of similarity and dissimilarity of metabolites of TEF at different time intervals. This tool provides an essential platform for rapid and vast information of metabolites for inferring the biological conclusions of TEF of *G. lucidum* [26].

GA-H based TEF of *G. lucidum* might be applied as an active phytopharmaceutical ingredient for the development of drug [22]. However, the problem associated with any phytopharmaceuticals development is its stability profiling. In this context, we have assessed the stability of TEF based on GA-H content. We found that TEF is stable for up to one year when stored at room temperature in both humid and non-humid conditions. This is the first time to report the assessment of triterpenoids stability in *G. lucidum*. This study gives a prominent clue for the quality, safety, and efficacy of GA-H of TEF for a longer duration of action.

From all aforesaid experiments, it was confirmed that standardized TEF contains well-known bioavailable phytochemical compounds. This standardized secondary metabolite enrichment fraction of *G. lucidum* can be explored for the development of phytopharmaceuticals.

## 4. Materials and Methods

### 4.1. Chemicals and Fruiting Bodies of Fungus

The analytical grade of GA-H (purity > 98.25%) was procured from LGC promochem (Bengaluru, India). LC–MS and HPLC grade solvents were procured from Merck Life Science (Mumbai, India). Solvents used for extraction, and other chemicals used were of analytical grade and procured from Super Religare Laboratories (SRL) limited (New Delhi, India). The fruiting bodies of *G. lucidum* were purchased from Aryan Mushrooms private limited (Ahmedabad, India) and authenticated as per the American Herbal Pharmacopoeia [27]. For future reference, sample specimens were stored at Bioactive Natural Product Laboratory (Voucher specimen No. JH/BNPL/*G. lucidum*/FB/2018).

### 4.2. Extraction, Fractionation, and Preparation of TEF

The dried fruiting bodies (1.0 kg) were pulverized and extracted three times with five liters of 95% ethanol, using the reflux method. The extract was concentrated under reduced pressure to yield dried mass, then suspended into 50 mL of hot water and extracted with the same volume of dichloromethane (DCM). The organic layer was concentrated to about 1/10th of its original volume and extracted with fifty milliliters of saturated aqueous sodium bicarbonate, and the extract was acidified to pH 3–4 with 6.0 mol/L HCl. The neutralized extract was stored at a cold temperature (0 °C) to become precipitated. The resulting precipitate was re-dissolved in DCM and then dried to get a triterpenoid enriched fraction (TEF). It was stored at −20 °C prior to analysis and other experiments.

### 4.3. Quantitative Estimation of GA-H

#### 4.3.1. Preparation of Standard Stock and Sample Solutions

Accurately weighed GA-H was dissolved in HPLC-grade methanol to get a final concentration of 1.0 mg/mL. Further, it was diluted with methanol to make different concentrations (100–1000 ng/mL) for linearity assessment. Similarly, TEF solution was also prepared in HPLC grade methanol (5.0 mg/mL), and it was filtered by using 0.45 μM filter (Millipore) before analysis.

#### 4.3.2. Instrumentation and Chromatographic Conditions

Alliance HPLC system (e2695 Separation module, Waters, Milford, MA, USA) with a gradient pump integrated with adjustable wavelength programmable photodiode array detector (PDA) was employed for analysis. Chromatographic separation was executed on Hupersil 5.0 µL C18 (ODS) column (150 × 4.6 mm, Phenomenex, Torrance, CA, USA), using mobile phase composed of 0.5% *v*/*v* formic acid in water (A) and methanol (B). Solvents were eluted at a flow rate of 1.0 mL/min in gradient elution program (initially 100% A, 0–5 min 90% A, 5–10 min 70% A, 10–15 min 30% A, 15–20 min 2% A, and 20–22 min 100% A). Detection was carried out in a scanning mode with 3D channel at a wavelength of 254 nm. Total run time was 25 min. The identity of the compounds was established by comparing retention time of analytical standards. The proposed developed method was further validated as per the AOAC guidelines [18].

#### 4.3.3. Validation of Developed HPLC Method

The chromatographic method for quantitative estimation of GA-H was validated for specific parameters, such as selectivity, linearity, sensitivity, accuracy, and precision, according to the Association of Official Analytical Chemists (AOAC) guidelines [19], in a similar manner as methods reported by our laboratory [28]. The linearity was analyzed through the standard plot of GA-H ranging from 100 to 1000 ng/mL. Different dilutions of standard were prepared by diluting stock solution (1.0 mg/mL) with methanol and analyzed in triplicate. The linearity was evaluated by linear regression equation plot analysis, which was calculated by the least-square regression analysis. Sensitivity of the developed method was measured by determining limit of detection (LOD) and limit of quantitation (LOQ). LOD (k = 3.3) and LOQ (k = 10) were calculated from the calibration curve, using *A = kσ*/*S* equation, where *A* is LOD or LOQ, *σ* is the standard deviation of response, and *S* is the slope of calibration curve.

The preciseness of the developed method was determined by measuring both intra-day and inter-day precisions. It was determined in six replicates of the mixed standard solution on the same day (intra-day precision) and daily for six times over a period of one week (inter-day precision). All results were expressed as % RSD for each parameter. The robustness of the developed method was studied with change in the mobile phase composition (methanol–water, 5:95, 10:90, and 15:85, *v*/*v*), temperature (25, 30, and 35 °C), and flow rate (0.9, 1.0, and 1.1 mL/min), respectively.

Accuracy of the method was measured by percentage recovery of GA-H after spiking it in two different matrices (plasma and crude extract), separately. For determination of recovery from plasma, standard solution of GA-H was spiked with plasma to get concentration of 500 and 1000 ng/mL. The spiked plasma was then extracted and analyzed to measure GA-H content. For extraction, 500 µL of HPLC grade methanol was added in 100 µL of spiked plasma. It was kept in refrigerator for 4 h for maximum protein precipitation and brought to room temperature. Further, it was centrifuged at 14,000 rpm for 10 min. The separated supernatant was dried by passing nitrogen gas and re-suspended in 100 µL of HPLC grade methanol. Similarly, standard solution was spiked in crude extract, and same extraction method was followed. The percentage recovery was calculated as the amount recovered with respect to spiked concentration.

### 4.4. Real-Time Stability Analysis of TEF

Real time stability assay of TEF was measured in terms of GA-H content. TEF was incubated in humid and normal conditions at two different temperatures (25 °C and 37 °C). Sampling was performed at three-month intervals up to one year. The collected samples were processed immediately and analyzed the GA-H content in TEF. Besides real-time stability assessment, TEF was exposed to ultra-violet radiation to know the effect of UV radiation on its stability and immediately analyzed for GA-H content [29,30].

### 4.5. Pharmacokinetic Analysis of TEF

#### 4.5.1. Animals and Treatment

Albino Wistar rats (8 weeks old, 180 ± 20 g) were obtained from the Central Animal House Facility of Jamia Hamdard. Animal housing and handling were executed in accordance with the Good Laboratory Practice (GLP) mentioned in CPCSEA guidelines. All experimental protocols were reviewed and approved by the Institutional Animal Ethics Committee (Registration No 173/GO/RE/S/2000/CPCSEA; approval number 1472). All animals were housed in polypropylene cages (not more than three animals per cage) with proper water and feed and were allowed to acclimatize with specified conditions (25 ± 3 °C and 60 ± 5%RH) for a week before the experiment. TEF at a dose of 85 mg/Kg was orally administered to overnight fasted rats. For oral administration, TEF was suspended in 0.1% carboxymethyl cellulose. After oral administration, blood samples were collected from retro-orbital plexus by using heparinized capillaries at 0 (pre-dose), 0.5, 1, 1.5, 2, 4, 6, 8, 10, 12, 16, 24, and 30 h. Blood samples were collected in EDTA tubes and centrifuged at 7000 rpm to get plasma. The separated plasma was stored at −60 °C until analysis. After oral administration of TEF, few animals were placed in metabolic cages, total urine was collected at 4, 12 and 24 h. The collected urine from each animal was pooled for analysis.

#### 4.5.2. Sample Preparation

Liquid–liquid extraction was used for extraction of GA-H from plasma. In 100 µL of plasma, 500 µL of ice-cold methanol was added, vortexed, and incubated at −60 °C for 8 h. After incubation, it was brought to room temperature and centrifuged at 10,000 rpm for 10 min. The supernatant was separated and dried by passing liquid N_2_ gas. The dried extract was re-suspended with 100 μL of LC–MS grade methanol, sonicated for 10 min, centrifuged at 14,000 rpm for 10 min, supernatant separated, and then transferred into sample vials having insert for analysis. Similar sample preparation protocol and storage conditions were used for urine through liquid–liquid extraction.

#### 4.5.3. Pharmacokinetic Calculation

The maximum plasma concentration (C_max_) and time to attain maximum concentration (T_max_) were directly assessed by plasma concentration–time graph. Other pharmacokinetic parameters, such as AUC, AUC_0–t_, AUC_0–∞_, half-life (t_1/2_), elimination rate constant (K_el_), and volume of distribution (V_d_), were calculated from plasma concentration vs. time profile response curve by non-compartmental analysis, using software Pkf Excel plug-in program [31].

### 4.6. In Vivo Pattern Recognition of TEF

For in vivo pattern recognition of TEF, it was orally administered to rats. After its oral administration, blood samples were collected at different time intervals. Prepared plasma samples of pharmacokinetic analysis (0.5, 1, 3, 6, 12, 24, 48, and 72 h) were used for this study. Samples were analyzed by ultra-performance liquid chromatography–mass spectrometry (UPLC–MS) analysis. The UPLC–MS was performed on Water’s ACQUITY UPLCTM system (Waters Corp., Milford, MA, USA) equipped with a binary solvent delivery system, an autosampler, column manager, and a tunable MS detector (Synapt; Waters, Manchester, UK). Chromatographic separation was performed on a Water’s ACQUITY UPLC^TM^ BEH C18 (100.0 × 2.1 mm × 1.7 μm) column. The mobile phase used for chromatographic separation was 0.05% (*v*/*v*) formic acid in water (A) and 0.05% (*v*/*v*) formic acid in acetonitrile (B). Solvent elution was started with 5% B (0 min), then it was gradually increased to 100% B in 30 min, maintained for another 10 min, and then brought back to initial phase. The flow rate of mobile phase was 0.5 mL/min, and column temperature was set at 25 °C throughout the run time. The mass spectrometry was performed on a quadrupole orthogonal acceleration time of flight tandem mass spectrometer (Waters Q-TOF, Manchester, UK). The nebulizer gas was set to 500 L/h, the cone gas was set to 50 L/h, and the source temperature was set to 100 °C. The capillary voltage was set to 3.0 kV, and the sample cone voltage was set to 40 kV. Argon was employed as the collision gas at a pressure of 5.3 × 10^−5^ Torr. The Q-TOF was operated in positive mode with 1.0 min scan time and 0.02 s inter-scan delay. The accurate mass and composition for the precursor ions and the fragment ions were calculated by using MassLynx v 4.1 software incorporated with the instrument. Compounds were tentatively identified from database by comparing their reported m/z values with experimental values. Focus was given to the mass ranges from 130 to 600 m/z for lanosterol type triterpenoids and general terpene hydrocarbon. The databases used for identification of compounds were MassBank and HMDB.

Metabolomic comparison was performed with respect to analysis of TEF. For that, dried TEF was dissolved in LC–MS grade methanol and analyzed with same conditions used for plasma sample analysis. A correlation was established by analyzing metabolites present in blood before and after TEF administration and metabolites present in TEF. This metabolomic analysis was based on metabolites absorbed in blood after oral administration.

### 4.7. GC–MS Profiling of TEF

Around 100 mg of extract was dissolved in 1.0 mL methanol and vortexed for half an hour. Then, 100 µL methanolic solution was silylated by adding 100 µL of a mixture of N-methyl-N-trimethylsilyltrifluoroacetamide (MSTFA) and trimethylchlorosilane (TMCS) in pyridine (22:13:65 *v*/*v*/*v*) and then incubated at 30 °C for 2 h. The sample was cooled down quickly and re-constituted with 100 µL of hexane. Gas chromatography–mass spectrometry was performed in Agilent 7890A (Agilent Technologies, Santa Clara, CA, USA), equipped CTC-PAL, and auto-sampler attached with an MS detector. Two microliters of derivatized sample was injected with a 10:1 split ratio onto a 30 m × 0.25 mm × 0.25 μm HP-5 MS column (5% diphenyl and 95% dimethyl polysiloxane). Chromatographic separation was performed as per the reported method of our laboratory [32]. Separated compounds were identified by matching the spectrum from NIST library.

### 4.8. Statistical Correlation of Metabolites Presents in TEF

Principal component analysis (PCA) is an influential statistical tool used for analysis, pattern identification in terms of similarity, and differences among different groups, as validated in our previous publications [32]. It can be assumed that the category and sets of metabolites with specified m/z value in TEF of *G. lucidum* before and after oral administration of blood at different time intervals may be different. By implementing this concept, UPLC–MS analysis data and HPLC data were processed for statistical analysis. Both sets of data were converted into a table in which compound presence and absence were normalized as 1 and 0, respectively. The extended statistical module XLSTAT 2014.5.03 software was used to perform for PCA. The similarity, dissimilarity, and maximizing separability of TEF and in plasma metabolites at different time intervals were observed across the four quadrants.

### 4.9. Statistical Analysis

GraphPad Prism version 8.0.2 (GraphPad Software, 2365 Northside Dr. Suite 560, San Diego, CA USA) software was used for the statistical analyses. Two-way analysis of variance (ANOVA) was used to analyze data, and the differences were analyzed by using a multiple comparison (Bonferroni’s test). The results were expressed as (% recovery ± SEM). Differences were considered significant if *p* < 0.05.

## 5. Conclusions

Triterpenoid enriched fraction (TEF) was prepared from the fruiting bodies of *G. lucidum* in which GA-H was quantified by a newly developed and validated HPLC method. Upon oral administration of TEF, the rapid absorption of GA-H into systemic circulation was observed, and it showed longer systemic exposure. Metabolomic fingerprinting revealed that, apart from GA-H, TEF contains other bioavailable ganoderic acids. Based on GA-H content, TEF is stable for up to one year in both humid and normal atmospheric conditions at room temperature. Thus, prepared TEF from Indian varieties of *G. lucidum* can be explored as a potential phytopharmaceutical for unmet medical needs.

## Figures and Tables

**Figure 1 metabolites-12-00097-f001:**
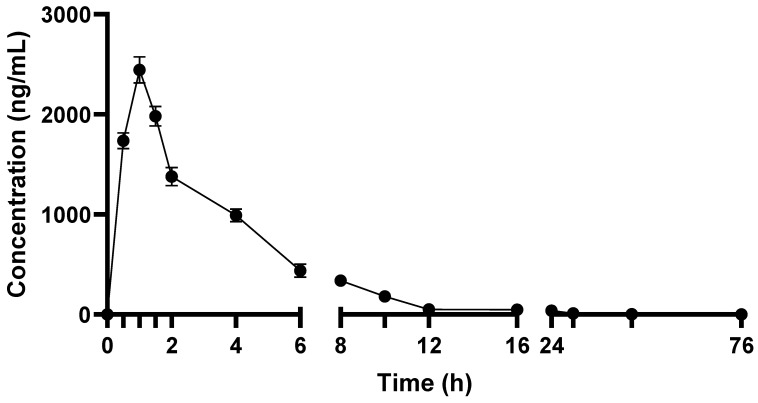
Plasma concentration–time profile of TEF in terms of GA-H after its oral administration to rats.

**Figure 2 metabolites-12-00097-f002:**
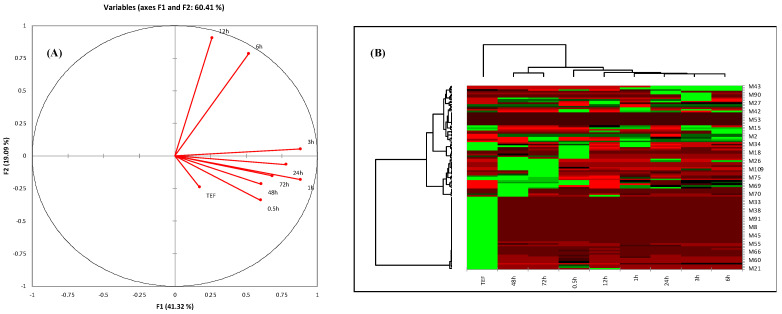
(**A**) PCA score plot of correlation circles showing the different collections of samples based on their metabolite pattern. Different quadrants show similarity of metabolites’ pattern and vice versa. (**B**) Heat map of analyzed metabolites obtained from HPLC and UHPLC–MS profiling. Row correspondence to abundant and metabolite causing differentiations, while column indicates metabolites abundance with respect to sample. Abundancy of metabolites is shown in different colors: red color represents the higher metabolites abundancy, green color represents lower abundancy, and black color represents very lower abundance of metabolites.

**Table 1 metabolites-12-00097-t001:** UPLC–MS based metabolic profiling of TEF and blood sample after its oral administration.

R_t_ (min)	Exact Mass	Theoretical [M + H]^+^	Experimental [M + H]^+^	Tentative Name	Formula	Class	Metabolites Pattern in TEF and Blood (Abundance)								
							TEF	0.5 h	1 h	3 h	6 h	12 h	24 h	48 h	72 h
0.99	-	-	158.2601	Unknown	-	-	20,897	ND	ND	ND	ND	ND	ND	ND	ND
2.75	226.1066	227.1139	227.0998	L-Carnosine (KNA00260)	C_9_H_14_N_4_O_3_	Peptide	17,546	ND	ND	ND	ND	ND	ND	ND	ND
3.63	452.1319	453.1391	453.4025	Epicatechin-6-glucoside (HMDB37399)	C_21_H_24_O_11_	Flavonoids	61,237	19,876	36,783	25,056	26,578	22,346	21,324	24,569	22,345
4.13	339.1066	340.1139	340.4901	Glutamine-betaxanthin (HMDB0304684)	C_14_H_17_N_3_O_7_	Amino acids	32,678	18,234	23,367	24,356	25,643	25,461	21,675	ND	ND
5.74	530.2879	531.2952	531.5023	Ganoderic acid N (HMDB0035325)	C_30_H_42_O_8_	Triterpenoids	98,785	ND	22,897	36,543	48,654	32,453	31,954	ND	ND
5.90	514.2931	515.3003	515.4392	Ganoderic acid D (14109406)	C_30_H_42_O_7_	Triterpenoids	75,453	ND	36,786	37,654	27,689	26,657	21,234	ND	ND
5.93	516.3087	517.3160	517.3564	Ganoderic acid A (471002)	C_30_H_44_O_7_	Triterpenoids	65,675	ND	45,647	37,652	29,143	22,987	21,548	ND	ND
5.96	498.3709	499.3782	499.3707	Tsugaric acid A (HMDB0032022)	C_32_H_50_O_4_	Triterpenoids	48,975	ND	ND	ND	ND	ND	ND	ND	ND
6.12	556.4128	557.4201	557.4801	Hericene A (HMDB0041179)	C_35_H_56_O_5_	Monoterpenoid	21,345	ND	ND	ND	ND	ND	ND	ND	ND
6.12	-	-	570.0512	Unknown	-	-	33,658	ND	ND	21,234	20,657	ND	ND	ND	ND
8.13	572.2985	573.3058	573.1841	Ganoderic acid H ^*^	C_32_H_44_O9	Triterpenoids	176,859	35,876	45,675	54,675	32,456	24,978	ND	ND	ND
9.06	281.1416	282.1489	282.4015	Coumarin 106 (BML01761)	C_18_H_19_NO_2_	Organic compounds	46,573	ND	23,145	21,222	19,873	8723	ND	ND	ND
9.27	242.2246	243.2319	243.4332	13-methylmyristic acid (RP024601)	C_15_H_30_O_2_	Fatty acids	56,435	ND	ND	23,412	25,473	21,675	16,734	9835	5642
11.24	280.2402	281.2475	281.3921	Linoleic acid (EQ331601)	C_18_H_32_O_2_	Fatty acids	56,874	ND	ND	23,451	21,543	18,756	12,112	8453	5641
Total metabolites: 14							14	03	07	10	10	09	07	03	03

Note: Values presented in tables are the abundances of respective masses. ND: not detected. Compound marked with * was verified with analytical standards.

**Table 2 metabolites-12-00097-t002:** GC–MS profiling of TEF and identified major compounds.

Compound Name (% Matching with NIST Library)	R_t_ (min)	Formula	Area	Area%
Tetradecanoic acid (91)	22.51	C_14_H_28_O_2_	1,828,311	0.43
n-Hexadecanoic acid (99)	26.6	C_16_H_30_O_2_	16,070,781	3.85
9,12-Octadecadienoic acid (Z,Z) (99)	29.83	C_18_H_32_O_2_	26,470,635	6.34
Octadec-9-enoic acid (99)	29.93	C_18_H_34_O_2_	22,152,301	5.31
Oleic acid, trimethylsilyl ester (97)	31.36	C_21_H_42_O_2_Si	4,329,461	1.03
3-benzyl-1,4-diaza-2,5-dioxobicycl-o [4.3.0]nonane Pyrrolo[1,2-a]pyrazine-1,4-dione (89)	34.04	C_10_H_14_N_2_O_3_	6,448,442	1.54
Hexadecanoic acid (91)	36.12	C_16_H_30_O_2_	28,726,695	6.88
Methyl 2-hydroxy-octadecanoate (80)	38.07	C_19_H_38_O_3_	10,624,044	2.54
Methyl 6,8-dodecadienyl ether (81)	38.22	C_13_H_24_O	9,457,891	2.26
2,6,10,14,18-Pentamethyl-2,6,10,14, 8-eicosapentaene (92)	40.81	C_30_H_50_	16,139,150	3.86
14.alpha.-Cheilanth-12-enic Methylester (80)	43.29	C_25_H_42_O_5_	1,966,254	0.47
*β*-Carotene (82)	44.9	C_40_H_56_	2,584,634	0.61
*α*-Gurjunene (87)	55.83	C_15_H_24_	1,990,618	0.47

**Table 3 metabolites-12-00097-t003:** Real-time stability analysis at 25 and 37 °C (non-humid and humid conditions at specified times 0, 3, 6, 9, and 12 months of TEF of *G. lucidum* in terms of GA-H (µg/mg) equivalent.

Time (Month)	TEF in Terms of GA-H Content µg/mg (Mean ± SEM)				
	25 °C		37 °C		
	Non-Humid	Humid	Non-Humid	Humid	
**0**	22.7 ± 1.3	22.6 ± 1.2	22.1 ± 2.6	21.5 ± 2.1	
**3**	22.1 ± 2.7 ^ns^	21.9 ± 2.3 ^ns^	21.5 ± 3.9 ^ns^	20.2 ± 0.7 ^ns^	
**6**	22.0 ± 4.1 ^ns^	21.3 ± 1.7 ^ns^	20.8 ± 4.9 ^ns^	19.7 ± 2.1 ^ns^	
**9**	21.4 ± 2.9 ^ns^	20.4 ± 3.4 ^ns^	20.0 ± 2.8 ^ns^	19.1 ± 2.2 ^ns^	
**12**	20.7 ± 1.84 ^ns^	20.3 ± 1.2 ^ns^	21.5 ± 1.5 ^ns^	18.9 ± 2.5 ^ns^	

Note: *p* > 0.05; data were compared with 0 month, ns represents non-significant.

**Table 4 metabolites-12-00097-t004:** Eigenvalues of different variables.

	F1	F2	F3	F4	F5	F6	F7	F8	F9
Eigenvalue	3.719	1.718	1.207	0.812	0.649	0.414	0.273	0.121	0.088
Variability (%)	41.321	19.091	13.407	9.022	7.208	4.597	3.032	1.344	0.977
Cumulative%	41.321	60.412	73.819	82.841	90.050	94.646	97.679	99.023	100.000

**Table 5 metabolites-12-00097-t005:** Correlation matrix of variables.

Variables	TEF	0.5 h	1 h	3 h	6 h	12 h	24 h	48 h	72 h
TEF	1	0.385	0.094	0.085	0.025	−0.036	0.038	0.029	0.040
0.5 h		1	0.498	0.464	0.148	−0.048	0.238	0.495	0.251
1 h			1	0.875	0.307	−0.006	0.667	0.442	0.530
3 h				1	0.483	0.206	0.620	0.342	0.495
6 h					1	0.807	0.238	0.157	0.194
12 h						1	0.141	0.043	0.018
24 h							1	0.412	0.623
48 h								1	0.343
72 h									1

## Data Availability

The data presented in this study are available in Appendix A.

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
