# Peer review of "Pharmacokinetic, Metabolomic, and Stability Assessment of Ganoderic Acid H Based Triterpenoid Enriched Fraction of Ganoderma lucidum P. Karst"

_metabolites, 2022, doi:10.3390/metabo12020097_

Round 1

Reviewer 1 Report

Despite the need and importance of such manuscript on natural products with in depth analysis, this manuscript has lot of flaws and significant amount of errors which needs further justification and corrections. Please see below few of my major comments:

  1. First and the most important is English language. Though, I started correcting all typo, syntax, punctuations, Grammatical errors, but at one point I gave up. This manuscript is good but poorly written. Language of the manuscript is obstructing the flow of reading and understanding at various points. Please do not revise the language by yourself, if you really want this manuscript to be sound. Ask any native English speaker or professional editing service to help you with it.
  2. Introduction needs further elaboration. Introduction in this manuscript feels like more of discussion. Start the introduction with natural product, it importance and significance, then move to Ganoderma lucidum and its chemistry, then move to the need of this study...etc....maintain the sequence. 
  3. Abstract needs complete refining. Look at your method section of abstract, you are discussing aims and goals in it. Conclusion is starting with Therefore.....
  4. It will be better if you can provide the scatterplot for your r2 value in section 2.1
  5. I need to see all the figures in high resolution again, especially supplementary figures. I cannot assess the figures and related data in the manuscript. Figures are of very very bad quality, when I am zooming to see the retention times, intensity and peaks, it gets all blurred. They are too small. No units and numbers can be seen at all.
  6. Table 2. I cannot see 5 degree celsius in the table. What is the reason behind in choosing these specific temperatures. It must be discussed. Stability can be further checked at more higher as well as lower temperature.
  7. In figure 3, what are these bar graphs representing. It must be clearly stated in legend. Moreover, it looks to me very high.
  8. In section 2.5, you said 14 metabolites were found. However in table it is 19. 
  9. It should be pelargonic acid and perlargonic acid.
  10. How many times you run the profiling? because I am not sure that you will find pelargonic acid, pentamidine, vecuronium etc. It is not possible.
  11. 8 were observed in systemic circulation...which 8???
  12. 5 were detected with other class.....which 5???
  13. Table 4 needs to completely rechecked. Especially molecular formula. They are not correct. I cannot match retention times due to poor figure quality. For example, Ganoderic acid C2 should be A
  14. Are you sure in table 4 that vecuronium is a terpenoid alkaloid?? I dont think so.
  15. Do you really think pentamidine and tauroursodeoxycholic acid will be present?
  16. Discussion, paragraph 2, line 13-20. All the results of these parameters should be submitted as supplementary file. furthermore, the same paragraph continues without any break in paragraph for two pages. As i told above, you are making very hard for the readers to understand this manuscript.

Reviewer 2 Report

The authors report on the biological properties of an enriched fraction in GA-H based triterpenoids of G. lucidum. The results are usually well described, but an improvement must be introduced for their presentation.

They must discuss the choice of using only one value of wavelength and 254 nm in their HPLC analysis.

Figures 1, ,2, 3 ,6 and figures in Supplementary are too small and the resolution can be increased. In tables 2 and 3 the deviations are very high, so that it is out of physical meaning the writing i.e. 22.72 ±3.28  which is instead 22±3 considering the significant digits of that number.

Regarding page 6, line 183, the identification of metabolites is based only on m/z value; why not considering the UV profile?

The columns in table 4 must be optimized and the suggestion is to present the molecular structures of the metabolites in an appropriate figure.

Experimental data on pharmacokinetic properties can be supported by comparison with predicted values for the 19 identified metabolites.

Minor comments:

uniformy ml as mL according to IS.

 At page 6 m/z must be reported at the left of each numerical value.

Author Response

Please see the attchment.

Reviewer 3 Report

The authors have carried out a pharmacokinetic study on the triterpenoid enriched fraction of an Indian sample of G. lucidum and its potential as a pharmaceuctical agent. The following revisions are needed for a satisfactory manuscript:

  1. Clear and detailed experimental procedures. For example 4.1, no information on location from which specimen was collected. 4.2, samples was extracted 3x- how much solvent? how much dried mass resulted? volume of H20? CH2Cl2? how much final dried TEF?
  2. Metabolite identification in Table 4 seems to be based solely on molecular ion which is unacceptable. There must be additional matching with UV/vis spectra and molecular fragments and I have no confidence in their assignment. For example, they report synthetic drugs such as pentamidine and vecuronium and the Streptomyces natural product cycloheximide which are highly unlikely to be present in Ganoderma.
  3.  There is no critical analysis of the findings. The data needs to be compared to results in the extensive Chinese literature on G. lucidum. Are the bioactive constituents in their sample similar in identity and quantity to other Ganoderma preparations? Are there significant differences?

Reviewer 4 Report

Dear Authors,

I have the following comments to your work:

  • your work needs to be corrected by a native speaker. It is not smooth and at some points not clear at all.
  • i have a serious comment to the hplc method that you treat as the optimized. it can be shown in the figure 2 that the peak of  GA-H is overlapping with other constituents of the injected sample. this poor separation makes all quantitative data incorrect. the LC method should be again prepared in a different gradient of the  solvents to show clearly separated peak of GA-H from the complex.

other comments:

  • in the introduction, please explain why the Authors are interested in this particular compound. Are there any records of its activity?
  • please, be more precise in the materials and methods section. in the chapter 4.2. describe in details why you are taking these particular steps (liquid-liquid separations, resuspending in water, in DCM etc.). was this procedure already published? it can give big loss in quantity...which temperature was used for evaporation?
  • please, clearly describe where the fruiting bodies of ganoderma lucidum were obtained from, who authenticated the sample, which laboratory has the voucher specimen
  • in the line 350 these are fruiting bodies and not fruits
  • why did the Authors use this particular concentration range 100-1600 ng/mL for a standard. why was it that low? please, comment on that
  • the section 4.3.2. should have all chrmatographic method parameters - the composition of gradient, temperature, UV detection details...please, move it from the results section

Round 2

Reviewer 1 Report

Though authors have done significant and extensive changes, I am sorry to say, that manuscript is in mess and very difficult to understand and go through. Unfortunately, I am unable to follow the manuscript due to huge amount of editing and track changes. Where the manuscript is going, I really cant judge from the current form of the manuscript. I would like to see the revised manuscript again after the acceptance of all these track changes and final sanitized copy, so I can read properly and pass my decision.

Furthermore, Table 3 from discussion must be moved to result section. Please add GC-MS section in results and explain the results. Similarly, obtained GC-MS results must be discussed in the discussion section.

Author Response

Comment 1: Though authors have done significant and extensive changes, I am sorry to say, that manuscript is in mess and very difficult to understand and go through. Unfortunately, I am unable to follow the manuscript due to huge amount of editing and track changes. Where the manuscript is going, I really can’t judge from the current form of the manuscript. I would like to see the revised manuscript again after the acceptance of all these track changes and final sanitized copy, so I can read properly and pass my decision.

Answer: We understand that the track changes manuscript is difficult to follow because it had significant changes. We have provided clean copy as well as highlighted track change copy. We wrote note to editorial members for forwarding clean copy and track change copy to reviewers.

Comment 2: Furthermore, Table 3 from discussion must be moved to result section. Please add GC-MS section in results and explain the results. Similarly, obtained GC-MS results must be discussed in the discussion section.

Answer: As suggested, table 3 has been moved to result section and appropriate information has been incorporated in discussion sections.

Reviewer 2 Report

In the final part of Introduction at page 3, the authors must better distinguish the state of the art and the aims of their work, describing the latter one at the end of the paragraph. i.e. “But in this study, we found GA-H is one of the major triterpenoids found in Indian varieties of G. lucidum” doesn't go there

In results, page 3 and 4 the workup at 80 °C for evaporating ethanol is dangerous because it can degrade metabolites, furthermore it is    not necessary if reduced pressure is applied because boiling points decrease in comparison with atm pressure.  

In Supplementary, the caption of Figure S6 must be completed.

Author Response

Comment 1: In the final part of Introduction at page 3, the authors must better distinguish the state of the art and the aims of their work, describing the latter one at the end of the paragraph. i.e. “But in this study, we found GA-H is one of the major triterpenoids found in Indian varieties of G. lucidum” doesn't go there

Answer: Thank you for the suggestion. Appropriate changes has been incorporated in revised manuscript.

Comment 2: In results, page 3 and 4 the workup at 80 °C for evaporating ethanol is dangerous because it can degrade metabolites, furthermore it is not necessary if reduced pressure is applied because boiling points decrease in comparison with atm pressure.  

Answer: Authors are thankful for critical evaluation. It was a typographical error. It has been corrected in revised manuscript.

Comment 3: In Supplementary, the caption of Figure S6 must be completed.

Answer: Again, we are thankful for pointing out mistakes. It has been corrected.

Reviewer 3 Report

There are many issues with the experimental data and lack of detail provided. Unless these are significantly improved, the manuscript must be rejected.

  • Metabolite identification is based on poor quality MS data from a Q-TOF instrument and is unreliable- see examples below. For metabolomic studies,  HRMS, UV-Vis absorption spectra and MS-MS fragmentation patterns must all match literature values or identity confirmed by the use of reference standards.
  • Table 1- relative area % missing, and no discussion on whether the  Indian sample differs in ganoderic acid content compared to previous studies.
  • Table 1- MS identification is unreliable. For example, ganoderic acid F M+H exact mass is 571.2907 observed 570.0512. For the observed mass, the top 50 predicted molecular formulae do not contain C32. Ganoderic acid H C32H44O9 = m/z 572.2985 (and +1 for M+H). Instead, observed m/z is 571.5101. Every entry needs to be carefully checked, and differences between observed and calculated mass that are beyond acceptable tolerance must be explained. The authors should refer to guidelines on unambiguous molecular formula identification - see https://publish.acs.org/publish/author_guidelines?coden=jamsef
  • Table 1- plasma metabolites that are not arising from a TEF constituent should be excluded from the analysis e.g. phenylalanine, lysophospholipid.
  • Table 3 - same issues as Table 1 with quality of data and metabolite identification. There are plasticizers like phthalic acid, isobutyl nonyl ester, a synthetic 2,4-dinintrophenylhydrazine etc. In many cases, the degree of matching with NIST is too low - anything below 80% should be discarded, as should be components present in trace amounts.

Overall, this manuscript is heavily dependent on mass spectrometry as an analytical technique, but suffers from poor or incorrect interpretation of the data.

Author Response

Comment 1: Metabolite identification is based on poor quality MS data from a Q-TOF instrument and is unreliable- see examples below. For metabolomic studies, HRMS, UV-Vis absorption spectra and MS-MS fragmentation patterns must all match literature values or identity confirmed by the use of reference standards.

Comment 2: Table 1- relative area % missing, and no discussion on whether the Indian sample differs in ganoderic acid content compared to previous studies.

Answer: Relative abundances of each compound have been mentioned in revised manuscript. As suggested, ganoderic acid content in Indian verities differ with other verities has been discussed.

Comment 3: Table 1- MS identification is unreliable. For example, ganoderic acid F M+H exact mass is 571.2907 observed 570.0512. For the observed mass, the top 50 predicted molecular formulae do not contain C32. Ganoderic acid H C32H44O9 = m/z 572.2985 (and +1 for M+H). Instead, observed m/z is 571.5101. Every entry needs to be carefully checked, and differences between observed and calculated mass that are beyond acceptable tolerance must be explained. The authors should refer to guidelines on unambiguous molecular formula identification - see https://publish.acs.org/publish/author_guidelines?coden=jamsef

Answer: Thank you for pointing out blunder mistake. It was author’s oversight. Ganoderic acid F has now been replaced as ‘12-Acetoxy-7-hydroxy-3,11,15-trixolanost-8,20-dien-26-oic acid’. Mass spectra of GA-H (m/z 573.18) have been incorporated in revised manuscript. However, table 1 has been rechecked and exact mass has been incorporated to make sure the right compounds has been mentioned.

Comment 4: Table 1- plasma metabolites that are not arising from a TEF constituent should be excluded from the analysis e.g. phenylalanine, lysophospholipid.

Answer: Our aim was to identify the pattern of metabolites present in TEF after its oral administration to rats. Level of some indigenous metabolites were changed upon TEF administration. This table reflect the major bioavailable and non-bioavailable metabolites present in TEF as well as the affected indigenous metabolites.

Comment 5: Table 3 - same issues as Table 1 with quality of data and metabolite identification. There are plasticizers like phthalic acid, isobutyl nonyl ester, a synthetic 2,4-dinintrophenylhydrazine etc. In many cases, the degree of matching with NIST is too low - anything below 80% should be discarded, as should be components present in trace amounts.

Answer: Authors appreciate the reviewer for critical evaluation. It was a nice suggestion. We have removed compound having <80% matching with NIST library in supplementary file as designated as unknown compounds.

Reviewer 4 Report

Dear Authors, I accept the obtained corrections

Author Response

Comment 1: English language and style are fine/minor spell check required

Answer: The language correction has been done as per your suggestion.

Round 3

Reviewer 1 Report

Manuscript now can be accepted in its current form. English editing is required, which I believe can be done at later stage.

Author Response

Thanks

Reviewer 3 Report

Despite comments and suggestions of the previous drafts, the quality of data in the revised manuscript is scientifically unacceptable.

  • UPLC-MS: The experimental data provided in the SI does not show resolution MS. The figures in Table 1 are not making sense  e.g. just taking the first entry hydroxyproline, exact mass for C7H11NO3 is 157.0739 but reported as 157.1690. Even with the incorrect figure, (M+H) is 158.0817 which is far from the reported figure of 158.2601. It would be impossible to assign hydroxyproline based on such numbers.
  • Table 1 assignment as +++, ++ or + is meaningless without explanation of the ranges associated with these abundances.
  • Table 1 is meaningless for endogenous metabolites not arising from TEF. Even if their concentration changes, proper controls of non-TEF treated plasma at various time points is necessary. Due to statistical errors, multiple replicates are necessary as well. 

Author Response

Subject: Third revision of a manuscript entitled “Pharmacokinetic, metabolomic, and stability assessment of ganodericacid H based triterpenoid enriched fraction of Ganoderma lucidum P. Karst” (Manuscript Number: metabolites-1423742).

Dear Ms. Lora Chen:

We are thankful to you and the reviewers for their valuable comments and recommendation on our submittedmanuscript, entitled, “Pharmacokinetic, metabolomic, and stability assessment of ganoderic acid H based triterpenoid enriched fraction of Ganoderma lucidum P. Karst, MS# metabolites- 1423742.” The reviewers suggested a minor revision in our manuscript. As suggested, we revised table 1 and its associated text in the revised manuscript. One of our senior Professors has checked this manuscript thoroughly for its grammatical, spelling, linguistic error, and corrections incorporated in the revised manuscript.

We hope that you will now consider the manuscript suitable to be a potential publication in metabolites.

Sincerely,

Pointwise comments and answer Reviewer 1

Comment 1: Manuscript now can be accepted in its current form. English editing is required, which I believe can be done at later stage.

Answer: Thank you. One of our senior Professors has checked this manuscript thoroughly for its grammatical, spelling, linguistic error, and corrections incorporated in the revised manuscript.

Reviewer 3

Comment 1: UPLC-MS: The experimental data provided in the SI does not show resolution MS. The figures in Table 1 are not making sense  e.g. just taking the first entry hydroxyproline, exact mass for C7H11NO3 is 157.0739 but reported as 157.1690. Even with the incorrect figure, (M+H) is 158.0817 which is far from the reported figure of 158.2601. It would be impossible to assign hydroxyproline based on such numbers.

Answer: Thank you for the suggestion. The above-said compound has been marked as unknown. Moreover, we never said that compounds were identified. Except, GA-H, which was confirmed through analytical standards, other metabolites were marked as ‘tentatively identified’. Throughout the manuscript, we never claimed that metabolites were identified. We have only suggested that these may be the metabolites. For confirmation, each metabolite should be compared with respective standards. We think this should not be a problem.

Comment 2: Table 1 assignment as +++, ++ or + is meaningless without explanation of the ranges associated with these abundances. Table 1 is meaningless for endogenous metabolites not arising from TEF. Even if their concentration changes, proper controls of non-TEF treated plasma at various time points is necessary. Due to statistical errors, multiple replicates are necessary as well. 

Answer: Authors are thankful for the suggestion. Relative abundances of respective masses have been incorporated in the revised manuscript.
